# Advanced Respiratory Models for Hazard Assessment of Nanomaterials—Performance of Mono-, Co- and Tricultures

**DOI:** 10.3390/nano12152609

**Published:** 2022-07-29

**Authors:** Laura Maria Azzurra Camassa, Elisabeth Elje, Espen Mariussen, Eleonora Marta Longhin, Maria Dusinska, Shan Zienolddiny-Narui, Elise Rundén-Pran

**Affiliations:** 1National Institute of Occupational Health in Norway, 0033 Oslo, Norway; laura.camassa@stami.no; 2NILU—Norwegian Institute for Air Research, 2027 Kjeller, Norway; eel@nilu.no (E.E.); espen.mariussen@fhi.no (E.M.); eml@nilu.no (E.M.L.); mdu@nilu.no (M.D.); 3Institute of Basic Medical Sciences, Department of Molecular Medicine, University of Oslo, 0372 Oslo, Norway; 4Norwegian Institute of Public Health, FHI, 0456 Oslo, Norway

**Keywords:** 3D lung model, air–liquid interface, nanotoxicology, NM-300K, tricultures

## Abstract

Advanced in vitro models are needed to support next-generation risk assessment (NGRA), moving from hazard assessment based mainly on animal studies to the application of new alternative methods (NAMs). Advanced models must be tested for hazard assessment of nanomaterials (NMs). The aim of this study was to perform an interlaboratory trial across two laboratories to test the robustness of and optimize a 3D lung model of human epithelial A549 cells cultivated at the air–liquid interface (ALI). Potential change in sensitivity in hazard identification when adding complexity, going from monocultures to co- and tricultures, was tested by including human endothelial cells EA.hy926 and differentiated monocytes dTHP-1. All models were exposed to NM-300K in an aerosol exposure system (VITROCELL^®^ cloud-chamber). Cyto- and genotoxicity were measured by AlamarBlue and comet assay. Cellular uptake was investigated with transmission electron microscopy. The models were characterized by confocal microscopy and barrier function tested. We demonstrated that this advanced lung model is applicable for hazard assessment of NMs. The results point to a change in sensitivity of the model by adding complexity and to the importance of detailed protocols for robustness and reproducibility of advanced in vitro models.

## 1. Introduction

The exponential rise in production and use of NMs is increasing the risk of human and environmental exposure. Upon inhalation of particulate and chemical environmental pollutants, the respiratory system is the first-line target for adverse health effects [1,2]. This is emphasizing the importance of developing and validating advanced respiratory models for hazard identification and characterization of NMs to ensure safe use and reduced concern for public health. Development of new alternative methods (NAMs), including advanced three-dimensional (3D) in vitro models, are needed for next-generation human hazard and risk assessment, moving from in vivo animal experiments towards in vitro testing and in silico modeling, in compliance with the 3R principle to reduce, replace and refine animal experiments and as part of an integrated approach to testing and assessment (IATA) for regulatory use of in vitro data.

A large fraction of the NMs (diameter ≤ 100 nm) has been shown to deposit in the alveolar region of the lungs [3,4]. The interaction of the reactive NMs with the large surface area of the lungs is of importance for cellular uptake, and thus for the local and systemic effects [5]. Both environmental and engineered NMs have been shown to induce adverse health effects, such as pulmonary, cardiovascular, neurological, and reproductive disorders [6,7,8]. A direct association between exposure to diesel exhaust particles (DEPs) in exhaust fumes and lung cancer has been demonstrated [9,10,11,12]. Engineered NMs, (e.g., metallic nanoparticles, quantum dots, carbon nanotubes) have also been shown to cause toxicity after inhalation exposure [13,14,15]. At the cellular level, NMs can provoke oxidative stress, inflammatory responses, and/or genotoxicity. These molecular events play key roles in the development of adverse health effects of NMs [16].

The potential hazardous health effect of inhaled NMs depends on the interaction of deposited nanoparticles at the lung surface with the cells of the air–blood barrier [3,17,18,19,20]. Most of the pulmonary in vitro studies have been conducted using cell lines in submerged conditions, which is far from the in vivo situation. Additionally, the cell medium can influence the status and the properties of the NMs [21,22,23]. Therefore, it is important to establish robust and optimized in vitro models closer to the real-life inhalation exposure situation that can give a more realistic judgment regarding the hazardous potential of compounds, including particulate matters and NMs [24].

Closer to the scenario in vivo is the exposure of cells at the air–liquid interface (ALI), where cells cultivated on porous membranes are in direct contact with the air on one side and cell culture media on the other [9,25,26,27,28]. The aim of this work was to test for potential change in sensitivity for toxic insults when going from mono- to tricultures, and to test the robustness of and optimize an advanced 3D lung model in combination with an aerosol exposure system (VITROCELL^®^ cloud-chamber), to feature the air–blood barrier in the alveoli of the lower respiratory tract by performing and an interlaboratory trial across two laboratories. This advanced respiratory model, closer to the in vivo situation, can be used to study pulmonary processes and responses and as a valuable tool for hazard identification and characterization of NMs and environmental pollutants in the lungs after inhalation exposure.

We included in the model several human cell lines relevant for lung exposure, to advance from standard 2Ds to 3D cell cultures. We compared the response and sensitivity of mono-, co-, and tricultures of human lung epithelial cells (A549), endothelial cells (EA.hy926), and human differentiated monocytes (dTHP-1), respectively, after exposure in a VITROCELL^®^ cloud-chamber, to the JRC repository silver NMs NM-300K. Moreover, we aimed to establish a harmonized protocol and experimental design by performing an interlaboratory trial across two laboratories to strengthen the robustness of this advanced respiratory model for future applications for toxicity testing of NMs exposure.

## 2. Materials and Methods

### 2.1. Cell Cultures

The same batch of the cells were used for the inter-laboratory comparison studies performed at Laboratory 1 (NILU) and Laboratory 2 (STAMI). The human alveolar type II lung epithelial A549 [29], monocytic THP-1 [30], and endothelial EA.hy926 cells [31] were purchased from the American Type Culture Collection (ATCC, Manassas, VA, USA). The cell lines were cultured in DMEM or RPMI supplemented with 9–10% fetal bovine serum (FBS, product no. 26140079, ThermoFisher Scientific, Oslo, Norway; FBS, ultra-low endotoxin product no: S009Y20008, Biowest, vwr, Oslo, Norway) and 1% penicillin/streptomycin (Pen-Strep, product no. 15070063, ThermoFisher Scientific) (Appendix A) and maintained in an incubator with humidified atmosphere at 5% CO_2_ and at 37 °C. A549 and EA.hy926 cells were cultured at a density of approximately 1.3 × 10^4^ cells/cm^2^ in vented cell culture flasks. The cells were sub-cultured twice a week by dry trypsinization (0.25% for 2–4 min at 37 °C). THP-1 cells were maintained in suspension at a density of 3–9 × 10^5^ cells/mL for a maximum of 6–8 weeks.

For differentiation of THP-1 from monocytes to macrophage-like cells, phorbol-12-myristat-13 acetate (PMA, Sigma-Aldrich, St. Louis, MI, USA, product no. P8139, EU) was prepared as a stock solution (1 mg/mL) in DMSO. The stock solution was diluted in Milli-Q (MQ) to 10 µg/mL, and aliquots were kept at −20 °C in the dark. PMA at 20 ng/mL was added to undifferentiated THP-1 cells for 3 days. Differentiated THP-1 cells (dTHP-1) were cultured for 48 h in RPMI complete medium. Mature macrophages were evaluated by upregulation of the marker CD11b [32] by confocal laser scanning microscopy (LSM) (Section 2.10). dTHP-1 cells were harvested by incubation with 5 mL Accutase^®^ (Merck, Rahway, NJ, USA) for 10–15 min. In some cases, cell scraping was necessary in addition to Accutase^®^ incubation to detach the cells. Accutase^®^ was neutralized by adding complete cell culture medium. The cell suspension was centrifuged at 200× *g* for 5 min and resuspended in fresh complete medium.

A549, EA.hy926, and THP-1 were used at passages (P) 2–25, 3–19, and 3–15, respectively. Detailed information on passage numbers for each cell type is shown in Appendix A. All cell lines were tested regularly for mycoplasma contamination.

### 2.2. Advanced In Vitro 3D Lung Models

Monocultures of A549, cocultures of A549 + EA.hy926, and tricultures of A549 + EA.hy926 + dTHP-1 (Figure 1) were seeded at specified concentrations (Appendix A) on inserts of polyethylene terephthalate (PET) with 1 μm pore size and with a surface area of 4.2 cm^2^ (Falcon, BD Biosciences, San Jose, CA, USA) or 4.5 cm^2^ (Millicell, Merck). The choice of inserts with 1 µm pore size was to allow for spatial interaction between the cell types and at the same time to create compartmentalization between the cells. Mono-, co-, and tricultures were cultured in submerged conditions in 6-well plates (Falcon, BD Biosciences) for 48 h to let the cells grow to confluency. The cultures were transferred to ALI conditions by removing the apical medium and placed in the incubator for 24 h before exposure to the VITROCELL^®^ cloud system.

#### 2.2.1. Monocultures of A549

A549 cells were seeded at the density of 1.1 × 10^5^ cells/cm^2^ on the apical side of the cell insert in a 6-well plate, with 1 mL medium at the apical side and 3 mL medium at the basolateral side. The cells were incubated for 48 h before the basolateral medium was replaced by 1.5 mL fresh culture medium, and the apical medium was carefully removed to place the cells in ALI conditions before exposure.

#### 2.2.2. Cocultures of A549 and EA.hy926

EA.hy926 cells were seeded at the density of 1.1 × 10^5^ cell/cm^2^ on the basolateral side of cell inserts and placed in the incubator for 4 h. The inserts were then turned and placed inside of a 6-well plate with 3 mL of medium/well. A549 cells were seeded on top of the membrane insert in 1 mL medium, as described for monocultures.

#### 2.2.3. Tricultures of A549, EA.hy926 and dTHP-1

Tricultures were prepared as described for cocultures, but with inclusion of dTHP-1 cells. After 48 h incubation of the cocultures, the apical medium was removed. dTHP-1 cells were seeded at the density of 1.1–2.2 × 10^5^ cell/cm^2^ (4.9–10 × 10^5^ cells/insert) in 1 mL medium on top of the A549 cell layer. The same number of cells was seeded for all inserts within each experiment. After 4 h incubation, the basolateral medium was replaced by 1.5 mL fresh medium, and the apical medium was carefully aspirated to remove non-attached dTHP-1 cells and to place the cells in ALI conditions for 24 h.

### 2.3. NM-300K Dispersion and Characterization

#### 2.3.1. NM-300K Nanoparticles (Ag NMs)

The JRC Repository NM NM-300K and its dispersant NM-300K DIS were purchased from Fraunhofer IME, Germany. NM-300K are engineered spherical silver NMs with pristine size < 20 nm [33] and were chosen as a reference NM based on our previous work, including the work in NanoREG, showing toxicity of these NMs [34,35,36]. NM-300K DIS is a colloidal dispersion medium of deionized water (85%) containing 7% stabilizing agent (ammonium nitrate) and 8% emulsifiers (4% Polyoxyethylene Glycerol Trioleate and 4% Polyoxyethylene Sorbitan Monolaurate, Tween 20). NM-300K has a nominal silver concentration of 10% (*w*/*w*) [33]. The same batch of NM-300K was used by both laboratories.

#### 2.3.2. NM-300K Dispersion

A stock dispersion of NM-300K at nominal concentration 10 mg Ag/mL was sonicated in a solution of 0.05% bovine serum albumin (BSA) and MQ water to avoid agglomeration of silver particles, following the NANOGENOTOX protocol [37] with modifications. The dispersion was sonicated on ice and thereafter kept on ice for 10 min before use. At Lab 1, the sonication was conducted using a Labsonic^®^P sonicator and a 3 mm probe (product no 853 5124, Sartorius Stedim Biotech, Göttingen, Germany) at 50% amplitude for 5 min (100% cycle), or with a Q500 sonicator and a 3 or 6 mm microtip probe (Qsonica L.L.C, Newtown, PA, USA) at 30–35% amplitude for 7–8 min. At Lab 2, the sonication was conducted using a 400-Watt Branson Sonifier S-450D (Branson Ultrasonis Corp., Danbury, CT, USA) equipped with a standard 13 mm disruptor horn (Model number: 101-147-037), for 5 min and 10% amplitude. The sonicators were calibrated to give a delivered energy of 390–1090 J/mL of NM dispersion. The dispersion medium NM-300K DIS solvent control was prepared using the same procedure as for stock dispersion of NM-300K. For cell exposure at the lowest concentration, NM-300K stock was diluted in phosphate-buffered saline (PBS) without CaCl_2_/MgCl_2_.

#### 2.3.3. Dynamic Light Scattering Analysis of NM-300K

The hydrodynamic size and size distribution of NM-300K stock dispersion and diluted dispersion (in PBS) was measured at both laboratories using dynamic light scattering (DLS). The stock and diluted dispersions were diluted 1:100 in ultrapure water, mixed by pipetting, transferred to a disposable cuvette (DTS0012), and placed in the Zetasizer. At Lab 1 a Zetasizer Ultra Red (Malvern Panalytical Ltd., Malvern, UK) was used, with 3–5 measurements with automatic number of sub-runs at a fixed measurements angle of 174.4°. At Lab 2 a Zetasizer Nano ZS (Malvern Panalytical Ltd.) was used, with 3 measurements with 11 sub-runs at a fixed measurements angle of 173°. Analysis was performed at 25 °C with 120 s equilibration time, automatic attenuation, and no pause between repeats. Data were processed in the ZS Explorer software (Lab 1) or Zetasizer Nano software (Lab 2), using general purpose model, refractive index 1.59, and absorption 0.01. Results were presented as Z-average (Z-ave) which is the intensity weighted mean hydrodynamic size of the ensemble collection of particles, the polydispersity index (PDI), and hydrodynamic diameter (by intensity) of individual peaks in the size distributions.

Zeta potential (ZP) was also measured. The dispersion was diluted at 1:100 in ultrapure water, transferred to a disposable folded capillary cell (DTS1070) pre-wetted with ethanol and water, and placed in the Zetasizer Ultra Red. The ZP was measured by mixed mode measurement phase analysis light scattering (M3-PALS) at 25 °C.

For testing the stability of the NMs in different buffer solutions (PBS with and without CaCl_2_/MgCl_2_, HBSS with and without CaCl_2_/MgCl_2_, DMEM D6046 with no supplements), the dispersion was diluted 1:10 in the different buffers and further diluted 1:10 in ultrapure water, before size analysis by DLS in Zetasizer Ultra Red as described above.

#### 2.3.4. Analysis of Total and Dissolved Ag

The concentration of total and dissolved Ag species in the NM-300K stock dispersion (10 mg/mL) was measured by inductively coupled plasma mass spectrometry (ICP-MS). Directly after preparation, 0.5 mL of the stock dispersion was transferred to an Eppendorf tube and stored at room temperature (RT) until further processing for analysis of total Ag content. In parallel, directly after preparation, 1 mL of the NM-300K stock dispersion was transferred to Amicon Ultra centrifugal filter (3 kDa) unit tubes (Millipore, product no UFC900324) [36,38]. The filter was preconditioned with ultrapure water at 3900 g for 30 min before use. To separate Ag-NMs and dissolved Ag species, the samples were centrifuged at 3900 g for 45 min. Ultrapure water was used as control. The dissolved Ag in the filtrate (<3 kDa fraction) was stored at RT until further processing (Section 2.6). The proportion of dissolved Ag was calculated by dividing the measured Ag in the ultracentrifuged samples by the measured total Ag in the stock dispersion. The total Ag was measured from 7 independent experiments, each with one stock dispersion (*n* = 7), and the dissolved fraction was measured from two independent experiments, each with one dissolved fraction (*n* = 2).

#### 2.3.5. Endotoxin Testing of NM-300K

The NM-300K and NM-300K DIS were tested for possible endotoxin content with two different methods: HEK293 colorimetric test (InvivoGen) and the Limus Amebocyte Lysate (LAL Kinetic QCL) test.

##### HEK293 Endotoxin Test

The human embryonic kidney (HEK) 293 cell line endotoxin colorimetric test is based on the ability of the HEK293 toll-like receptor (TLR) 2 and TLR 4 transfected cells to recognize lipoteichoic acid (LTA) and lipopolysaccharide (LPS), respectively, from gram-negative bacteria (lipid A). These cells are engineered to be extremely sensitive to TLR-receptor agonists and further activation of the NF-κB pathway. HEK293 hTRL2 and hTLR4 cells co-express the NF-κB-inducible reporter gene secreted embryonic alkaline phosphatase (SEAP). The presence of the agonists LPS or LTA, starting as low as 0.03 ng/mL, will activate the HEK293 TLR2 and TLR4 receptors, respectively, and further the NF-κB pathway. NF-κB activation can be quantified using the dye HEK-Blue™Detection and reading the absorbance (OD) at 650 nm. The measured absorbance is directly proportional to the endotoxin concentration in the solution, where one endotoxin unit/mL (EU/mL) equals approximately 0.1 ng endotoxin/mL of solution.

Cells and reagents for the HEK293 endotoxin colorimetric test were purchased from InvivoGen. HEK293 hTLRnull, hTLR2, and hTLR4 cells were grown and maintained in DMEM high glucose, 10% FBS, and the antibiotics 10U penicillin/streptomycin (Gibco, Waltham, MA, USA) and Normocin (InvivoGen). Zeocin (InvivoGen) was additionally added to the cell medium for HEK293 hTLRnull, and HEK-Blue selection antibiotics (InvivoGen) were added to the cell medium for the maintenance of HEK293 hTRL2 and hTRL4 cells. HEK293 hTLRnull, hTLR2, and hTLR4 cell lines were exposed to NM-300K at concentrations of 1 µg/mL and 10 µg/mL for 24 h. Four replica exposures were conducted for each cell line. Cell viability was examined with the AlamarBlue assay as described in Appendix A. The absorbance was measured using a Spectrometer Gen 5 microplate data acquisition system and quantified using the analysis software (BioTek, Winooski, VT, USA). HEK293 hTLRnull cells were used as a negative control.

##### Limulus Amebocyte Lysate (LAL Kinetic QCL) Endotoxin Test

The LAL chromogenic endotoxin test (Lonza, EU) is based on a reaction between gram-negative bacterial endotoxin present in the analysis sample and a pro-enzyme, pro-factor C. The activation of the enzyme releases p-nitroaniline (pNA) from a synthetic substrate, producing a yellow color. The time required before the appearance of the yellow color is inversely proportional to the amount of endotoxin present. The yellow color from pNA is measured photometrically at 405 nm throughout the incubation period (30 min). NM-300K was tested for endotoxin with the LAL test at Lab 2, at the concentration of 0.5 and 50 µg/mL following the kit protocol. The concentration of endotoxin was calculated from its reaction time by comparison to a standard curve made by *E. coli* endotoxin. The LAL colorimetric test is extremely sensitive and detects as little as 0.1 EU/mL (approx. 0.01 ng endotoxin per mL).

### 2.4. Cell Exposure

The commercially available VITROCELL^®^ 6 Cloud System (VITROCELL^®^ Systems GmbH, Waldkirch, Germany) was used to expose the cells to NM-300K and controls. The system is equipped with an Aerogen Pro^®^ vibrating membrane nebulizer, which generates a dense cloud of droplets with a median aerodynamic diameter of 4–6 μm that are deposited at the bottom of the exposure chamber (area 145 cm^2^) [39] and is maintained at 37 °C in a laminar flow hood.

Transwell inserts with mono-, co- or tricultures were transferred to the VITROCELL^®^ 6 Cloud system, which was filled with an 18 mL cell culture medium/well to let the basolateral side of the insert be in contact with medium. A total volume of 300 µL (2 × 150 µL) of 10 mg/mL and 1 mg/mL of NM-300K dispersion was nebulized, to obtain, respectively, a nominal deposition concentration of 10 µg/cm^2^ and 1 µg/cm^2^ for exposure of the cells. See Appendix A for calculations on nominal deposition concentration. The same volume was used for exposure with the negative control PBS w/o CaCl_2_/MgCl_2_ and the solvent control NM-300K DIS. Exposures were performed in the same order in each experiment (PBS, NM-300K DIS, NM-300K 1 mg/mL, NM-300K 10 mg/mL) to avoid cross-over of solutions to the samples. Between each exposure, the nebulizer was rinsed with PBS, and the outlet and the box were wiped with a tissue. The plate was also wiped with tissue with ethanol between each exposure. The same nebulizer was used for all experiments at each laboratory, and was rinsed with PBS for several minutes, and wiped with a tissue between experiments.

For experiments performed at Lab 1, the nebulizer was rinsed with 150 µL PBS directly after the exposure to make sure a minimum of the NM dispersion was left in the nebulizer. Thus, the cultures were exposed to total volume of 300 µL sample and 150 µL PBS.

The cloud was allowed to settle (5–8 min) before the box was opened and the inserts transferred to new plates with a 1.5 mL fresh culture medium. Unexposed cultures (incubator control) were also transferred to new plate with fresh culture medium. The cultures were placed in the incubator for 24 h before processing for further analysis.

To defy the suitable control buffer in the experiment, cell cultures were exposed to several aerosolized buffer solutions. The buffer solutions of interest were PBS without CaCl_2_/MgCl_2_ (D8537 Sigma), PBS with CaCl_2_/MgCl_2_ (D8662 Sigma), Hank’s balanced salt solution (HBSS) without CaCl_2_/MgCl_2_ (14175-046 Gibco), HBSS with CaCl_2_/MgCl_2_ (14025-050 Gibco), and DMEM D6046 without supplements. Exposure of cell inserts to buffer solutions was performed using 3 × 150 µL buffer solution. The cloud settled for 8 min before the box was opened and the inserts were transferred to new culture plates with 1.5 mL fresh culture medium. The cultures were placed in the incubator for 24 h before processed for further analysis. Between exposures, the nebulizer was rinsed using the next buffer to test, with the order PBS without CaCl_2_/MgCl_2_, PBS with CaCl_2_/MgCl_2_, HBSS without CaCl_2_/MgCl_2_, HBSS with CaCl_2_/MgCl_2_, and DMEM without serum.

### 2.5. Deposition Efficiency and Barrier Integrity

#### 2.5.1. Fluorescein Measurements

To measure deposition efficiency in the VITROCELL^®^ cloud system, a fluorescent water-soluble fluorescein sodium salt (product no. 46960, CAS 518-47-8, BioReagent, Sigma-Aldrich) was used. At Lab 1, transwell inserts with 1 mL PBS were exposed to 150 µL aerosolized fluorescein (10 µg/mL). Aliquots of the PBS-fluorescein solution were transferred to a black 96-well plate for reading of fluorescence in a microplate reader at excitation 480 nm and emission 525 nm. The amount of fluorescein in each sample was quantified from a seven-point standard curve (0.625–100 ng/mL). Background fluorescence from the cell culture medium was subtracted from the measurements. The deposited fluorescein concentration per area was calculated by dividing the total amount of fluorescein in each sample by the area of the insert. The deposition efficiency of fluorescein in the VITROCELL^®^ system was calculated by dividing the deposited fluorescein (µg/cm^2^) by the maximum deposition per total area (1.5 µg/145 cm^2^), multiplied by 100%. This protocol for determination of deposition efficiency was similar to the VITROCELL^®^ protocol [40,41,42] that was used at Lab 2, where 200 µL of 15 µg/mL fluorescein in PBS was applied.

In parallel with deposition efficiency analysis, the cellular barrier integrity was investigated by measuring the break-through of fluorescein to the basolateral side of the cocultures. Cocultures of A549 and EA.hy926 were exposed at ALI to 150 µL aerosolized fluorescein (10 µg/mL). After incubation overnight, aliquots of the cell culture medium from the basolateral compartments were transferred to a black 96-well plate, with a standard curve, for fluorescence reading, as described above. Background fluorescence from the cell culture medium was subtracted from the measurements. The proportion of fluorescein transferred to the basolateral side of the membrane was calculated by dividing the amount recovered in the cell culture medium at the basolateral side of the membrane by the total deposition of fluorescein.

The deposition efficiency of the nebulizer used in the experiment was measured with 3 independent experiments, and the results were used to determine the required volumes and concentrations of NM-300K to obtain the nominal concentrations.

#### 2.5.2. ICP-MS Analysis of Ag

The amount of NM-300K deposited on the cells (mono- and cocultures) was quantified by ICP-MS to obtain precise information on the Ag amount delivered to the cells, and to determine the amount of Ag crossing the cellular barriers during exposure.

For deposition measurements, the well of the VITROCELL^®^ plate was filled with 10 mL PBS, ensuring no contact between the insert and the liquid, but keeping the system humid. NM-300K was nebulized as explained above for exposure of cells (Section 2.4) onto cell-free inserts, which after exposure were dried overnight in a 6-well plate (RT in the dark). The next day, the porous membrane filter was removed from the insert walls using scalpel and tweezer and transferred to a 5 mL Eppendorf tube. Blank, unexposed filters were included as control. The filters were stored in dark conditions at RT before further processing (Section 2.6). Deposition of NM-300K was calculated by dividing the total Ag amounts per filter by the membrane area. For the lower concentration, 2 independent experiments with each 2 inserts were used for the lower concentration, and 4 independent experiments with each 2–3 inserts for the higher.

The amount of Ag crossing the cellular barrier was also measured by collecting the basolateral medium into an Eppendorf tube, which was stored at RT before further processing. Results were calculated from 3 independent experiments (*n* = 3) each with 1–2 inserts (except for monocultures lower concentration where *n* = 2 experiments). Medium was also collected from dry, cell-free inserts with deposited NM-300K, incubated with 1.5 mL medium (DMEM D6046) below the insert for 24 h, to determine the maximum transfer of Ag through the insert (*n* = 2 independent experiments each with 1 insert). The relative amount of Ag crossing the cellular barrier was calculated by dividing the transferred Ag amount by the total deposited Ag.

### 2.6. Elemental Analysis of NM-300K by ICP-MS

The amount of Ag was measured by ICP-MS in liquid solutions (NM dispersions, water, cell culture media) and in filters from transwell inserts. Liquid samples were vortexed vigorously for at least 10 s before use. Approximately 0.25 g solution or complete filters were transferred to a Teflon container (18 mL) and subjected to microwave-assisted digestion with concentrated ultrapure distilled nitric acid mixed with ultrapure deionized MQ-water (2 mL water and 1 mL nitric acid). The samples were digested in an UltraCLAVE single reaction chamber microwave oven (Milestone, Italy) according to a 60 min stepwise heating program, with a hold time for maximum temperature (250 °C) at 15 min. The samples were allowed to cool down to RT in their vessels after digestion, before being transferred to 10 mL test tubes (VWR, polycarbonate) and diluted with deionized ultrapure water to a final volume of approximately 10 mL. Two blank samples, containing only ultrapure water and nitric acid, and one reference material were included in each digestion run. The reference material Oyster Tissue (1566b) from National Institute of Standards and Technology (NIST) containing 0.666 ± 0.009 µg/g Ag, was subjected to similar microwave-assisted digestion to assess the recovery of Ag. Mean recovery of Ag in the NIST Oyster Tissue was 0.642 ± 0.010 µg/g. The samples were analyzed for ^107^Ag by ICP-MS type Agilent 7700x (Agilent, Santa Clara, CA, USA), using the method accredited according to requirements of NS-EN/IEC 17025 (NILU-U-110).

### 2.7. Cytotoxicity Testing by the AlamarBlue Assay

Cytotoxicity was measured by the colorimetric assay AlamarBlue (AB) after 22–24 h exposure. This fluorometric assay is based on measuring cell viability by cellular reduction of the cell permeable, non-toxic dye resazurin into fluorescent resorufin in metabolically active cells. The fluorescence intensity is proportional to the number of living cells. The inserts were added to medium containing 10% *v*/*v* AB solution (Sigma-Aldrich) (1 mL on apical side, 1.5 mL on basolateral side) and incubated for 1–1.5 h. The plates were swirled gently to ensure proper mixing of the solution before aliquots of 40 µL or 100 µL (constant volumes within experiment) were taken from both compartments and transferred into a 96-well plate to measure the fluorescence intensity (ex.530 nm, em.590 nm). Blank values (medium with 10% *v*/*v* AB solution without cells) were subtracted from the measured fluorescence intensities. Cell viability was measured relative to unexposed incubator control (set to 100%). A minimum number of 3 independent experiments were performed with single or duplicate cell cultures. The inserts were not washed with PBS after exposure before addition of AB-medium, to minimize the loss of damaged/dead cells in the sample.

Control for interference between the NM-300K and read-out of the assay was included: cell-free inserts exposed to NM-300K were added 1 mL medium containing 10% *v*/*v* AB solution on the apical side and incubated for 1–1.5 h. Aliquots for fluorescence reading were performed as described above. Fluorescence intensity was compared to blank values (AB solution without NMs), and no differences were found (results not shown).

### 2.8. Genotoxicity Testing by the Comet Assay

The miniaturized 12-gel enzyme-modified version of the comet assay was performed to determine DNA damage (strand breaks, SBs) and oxidized base lesions, as described previously [35,43,44]. The first step was to harvest/collect the cells from the cell culture inserts. Directly after performing the AB assay, the cell cultures were washed twice with PBS (1 mL on apical side, 2 mL on basolateral side). The cells were wet trypsinized for 3–5 min in the incubator. For monocultures, 200–300 µL trypsin (0.25%, Sigma) was added to the apical side (dry basolateral side) and gently mixed after incubation to ensure proper disaggregation of the cell layer, before addition of 1 mL cell culture media for neutralization. For cocultures and tricultures, 200–300 µL trypsin (0.25%) was used on the apical side, while 1.5 mL trypsin (0.05%, Sigma) was used on the basolateral side of the membrane. The trypsin on the apical side was neutralized by 1 mL medium and on the basolateral side with 3 mL. The cells were resuspended by gently pipetting and transferred to Eppendorf tubes.

The cell suspensions were diluted in cell culture media to give approximately 200.000 cells/mL. Aliquots of the cell suspension were mixed 1:4 with low melting point agarose (0.8% *w*/*v*, Sigma-Aldrich, 37 °C) to a final agarose concentration of 0.64% *w*/*v*. Minigels (10 µL) with approximately 400 cells were made on cooled microscopic slides pre-coated with 0.5% standard melting point agarose (Sigma-Aldrich), with a maximum of 12 gels per slide. Slides were placed in Coplin jars and submerged in lysis solution (2.5 M NaCl, 0.1 M EDTA, 10 mM Tris, 1% *v*/*v* Triton X-100, pH 10, 4 °C) for 1–3 days. As a positive control for DNA strand breaks (SBs), separate slides were submerged in 100 µM H_2_O_2_ (in PBS, 4 °C) for 5 min, rinsed twice with PBS for 2 min, and then submerged in a separate Coplin jar with lysis solution.

For detection of oxidized or alkylated bases, the modified comet assay was used with the bacterial repair enzyme formamidopyrimidine DNA glycosylase (Fpg, gift from NorGenoTech, Oslo, Norway), which converts oxidized or alkylated bases to SBs [45]. After lysis, separate slides with cells embedded in gels were washed twice for 8 min in buffer F (40 mM HEPES, 0.1 M KCl, 0.5 mM EDTA, 0.2 mg/mL BSA, pH 8, 4 °C), added Fpg diluted in buffer F (final 60.000 × dilution), and covered with a polyethylene foil before incubation at 37 °C for 30 min in a humid box. Positive control for function of Fpg was performed regularly in the laboratory, by using a photosensitizer Ro 19-8022 (kindly provided by Hoffmann La Roche, Switzerland) with light to induce oxidized purines, mainly 8-oxoG, which is detected by the Fpg [46,47,48]. A549 cells were exposed to the photosensitizer Ro 19–8022 (2 µM) and irradiated with visible light (30 cm distance from cells, 250 W) on ice for 4 min, before embedding into gels. The positive control gave expected response based on historical controls (>20% DNA in tail for net Fpg).

The slides were placed in a horizontal tank submerged in electrophoresis solution (0.3 M NaOH, 1 mM EDTA, pH > 13, 4 °C) to let the DNA unwind for 20 min. Electrophoresis was run for 20 min at 25 V (1.25 V/cm, around 350 mA, Consort EV202, 4 °C). The gels were neutralized for 5 min in PBS and dH_2_O and dried horizontally overnight.

For quantification of DNA SBs, the gels were stained with SYBR gold (1:2000, Sigma-Aldrich), covered with a coverslip, and imaged in Leica DMI 6000 B (Leica Microsystems) equipped with a SYBR^®^photographic filter (Thermo Fischer Scientific). Comets of relaxed loops of DNA, withdrawn from the nuclei when subjection of DNA to the electrophoretic field after introducing breaks in the DNA, were scored using the software Comet assay IV 4.3.1 (Perceptive Instruments, Bury St Edmunds, UK). Median DNA tail intensity, proportional to the number of SBs, was calculated from 50 comets per gel as a measure of DNA SBs. Medians were averaged from 2–6 gels per cell culture. A total of 3–5 independent experiments were performed with single or duplicate exposure wells per experiment.

Control for possible interference between the NM-300K and analysis of comets was included. A sample of A549 control cells was mixed directly with NM-300K suspended in cell culture medium (140 µg/mL), just before embedding with LMP-agarose. The slides were handled in parallel with the other slides, as described above. The results from A549 cells analyzed with NM-300K present were compared to A549 control cells.

### 2.9. Uptake Analysis by Transmission Electron Microscopy

Engulfing of NM-300K by cells on the apical side of the insert (A549 and macrophage-like cells) was investigated by transmission electron microscopy (TEM). At 24 h after ALI exposure with 0.5 µg/cm^2^ NM-300K, the cells were washed twice with PBS, fixed with 2.5% glutaraldehyde in PBS for 20 min, washed with PBS, and postfixed with 1% OsO_4_ solution for 1 h. The cells were dehydrated in a series of ethanol with increasing concentrations and embedded in Epon–Durcupan resin. After polymerization for 72 h at 56 °C, ultrathin sections were cut at 70–80 nm using a Leica Reichert Ultracut Ultratome and Diatome knife and collected on 200 mesh size copper grids. Sections were contrasted with 2% uranyl acetate for 4 min and rinsed in MQ water. Contrast in lead citrate was avoided not to create deposition and interference during the microscopical investigation. Sections were examined with a Tecnai-12 transmission electron microscope operated at 80 kV.

### 2.10. Immunofluorescence and Confocal Microscopy

#### 2.10.1. Antibodies Staining

At 24 h postexposure, unexposed and ALI exposed tricultures were washed in PBS and fixed with a solution of 4% paraformaldehyde (PFA) in PBS for 15 min at RT. The membranes were removed from the plastic holder with a blade and kept in a 1:10 solution of fixative and PBS. The cells were permeabilized with a solution of 0.01% Triton-X 100 in PBS for 10 min. Unspecific epitopes were saturated in 2% BSA in PBS for 30 min. Cells were incubated overnight in a humidified chamber with primary antibodies directed toward cell markers specific to the air–blood barrier (Appendix A). The primary antibodies were diluted in 2% BSA in PBS. The membranes were washed in PBS and incubated with secondary antibodies conjugated to a fluorescent probe (Appendix A). Cells were stained with the nuclear staining DAPI for 5 min, washed in PBS and mounted on a microscopy slide with a mounting medium solution (Invitrogen), and covered with a coverslip. The cells were examined with a confocal Zeiss L-10 inverted microscope.

#### 2.10.2. Staining with Live Cell Markers

To identify differentiated THP-1 cells in tricultures, dTHP-1 cells were stained with CellTracker Green CMFDA (CTG, Invitrogen) which is a non-toxic dye that remains in the cell for 3–6 generations. dTHP-1 cells were incubated with 10 µM CTG in serum-free medium for 30 min, washed twice with PBS, before detached and seeded on top of the A549 cells on the apical side of the transwell insert. For tricultures, plasma membranes were stained with CellMask Deep Red Plasma Membrane Stain (Invitrogen), 1:750 dilution in serum-free medium for 15 min at 37 °C.

The cells were then washed in PBS and fixed in 4% formaldehyde for 15 min at RT. The transwell inserts were rinsed again with PBS and processed by carefully detaching the membrane with cells from the plastic walls. The membranes were then mounted between two glass coverslips with the mounting medium ProLong Gold Antifade Reagent with DAPI (Cell signaling Technology) for nuclei counterstaining. The samples were left to dry overnight at RT in the dark and later stored at 4 °C in the dark. Confocal microscopy was performed using a Zeiss LSM 700 (lasers 405, 488, and 639 nm; objective 40x). Image acquisition and processing were performed with the Zeiss Software ZEN. Z-stack acquisition was performed with 27–44 µm thickness, with 44 images for each stack.

The cell densities of A549 and dTHP-1 were estimated by manually counting the number of cells per image and dividing with the image area. The densities were compared to the seeding densities, and the A549 to dTHP-1 ratio was also calculated. A549 analysis was performed on 6 samples with a total of 25 images (2–10 images/sample) within three independent experiments (*n* = 3). dTHP-1 analysis was performed on 4 samples with a total of 17 images (2–10 images/sample) within two independent experiments (*n* = 2), with seeding densities 0.82–1.67 × 10^5^ cells/cm^2^. EA.hy926 cells were not counted.

### 2.11. Statistical Analysis

Results are presented as mean with standard deviation (SD) of at least 3 independent experiments (*n* = 3) with 1–2 replica inserts unless otherwise stated. Statistical analysis of AB and comet assay results was performed by comparing the mean of each sample to the mean of negative control (inserts exposed to PBS), by one-way ANOVA with multiple comparisons and post-test Dunnet using GraphPad Prism version 9.3.1 for Windows, GraphPad Software, San Diego, CA, USA. Level of significance was set to *p* < 0.05.

## 3. Results

### 3.1. Characterization of the Advanced 3D Lung Model

The orthogonal confocal pictures of the triculture model (Figure 2) show A549 (Figure 2a,c–e) and EA.hy926 cells (Figure 2b) growing at opposite sides of a transwell transparent membrane inserts, with cell membranes in red stained with cell mask red and nuclei in blue stained with DAPI. The A549 and EA.hy926 cells were evenly distributed on the membranes; however, some spots with fewer cells were also observed. A549 had a density of about 3.8 × 10^5^ cells/cm^2^ (SD: 0.3 × 10^5^ cells/cm^2^, *n* = 3). The dTHP-1 cells were located on the apical side of the transwell insert (in green), on top of the A549 cells. The dTHP-1 cells had a variable density and morphology (Figure 2c–e). Approximately 30% of the dTHP-1 cells seemed to be necrotic or damaged with irregular structures, although this was highly variable between images and samples. The proportion of dTHP-1 cells adhering and remaining in the culture compared to the seeded number of cells was estimated to be 25–38% (*n* = 2) by microscopic evaluation. The ratio between A549 and dTHP-1 cells in the tricultures was estimated to be 8–39 A549 cells per dTHP-1 cell (*n* = 3).

TEM analyses of tricultures showed cells compartmentalized in the apical and in the basolateral side of the insert with a 1 µm pore size (Figure 3a). EA.hy926 endothelial cells were localized at the bottom of the cell insert and the alveolar type II A549 cell on the upper side (Figure 3a). Cells that resemble the round morphology of THP-1 were seen on top of the epithelial A549 cells (Figure 3b), which were recognized by the presence of small microvilli (arrows) and lamellar bodies (LM). To further characterize the triculture lung model, dTHP-1 cells were stained with the mature macrophage marker CD11b (Figure 3c). The formation of tight junctions between A549 cells was visualized by labeling of ZO-1 protein in the cell membrane using LSM (Figure 3c). Distorted ZO-1 protein was found in the A549 cytoplasm (Figure 3c).

### 3.2. Characterization of AgNM-300K

The stock dispersions were characterized by Ag content (total and <3 kDa fraction), endotoxin content, hydrodynamic diameter, and zeta potential (Appendix A). The concentration of Ag in the stock dispersion was measured by ICP-MS to be 7.2 mg/mL ± 0.9 mg/mL (*n* = 7), which was lower than the expected nominal concentration of 10 mg/mL. A small amount of Ag (1.9 ng/mL ± 1.3 ng/mL, *n* = 3) was measured also in the dispersant control (NM-300K DIS). The amount of dissolved Ag in the dispersion, defined as <3 kDa fraction, was measured to be about 3.6% (*n* = 2). The NM-300K and NM-300K DIS stock dispersions were confirmed to be endotoxin-free by both HEK293 and LAL assays, being <1 EU/mL and <0.1 EU/mL, respectively (Appendix A).

The hydrodynamic diameter of NM-300K, measured by DLS, was found to be polydisperse with a Z-ave of 130.7 nm ± 23.2 nm with PDI of 0.380 ± 0.037 measured at Lab 1, and a Z-ave of 57.5 nm ± 5.7 nm with PDI of 0.343 ± 0.067 measured at Lab 2 (Appendix A). The hydrodynamic size of the NMs measured by Lab 1 was in general higher than at Lab 2, while the PDI was nearly the same in the dispersions from both laboratories, indicating a moderately polydisperse distribution type. The dispersions showed 2–3 peaks, where about 95% of the particles (by intensity) were in the peak of 202 nm ± 39 nm (Lab 1) or 89 nm ± 13 nm (Lab 2). The other peaks, with less intensity, were similar between both laboratories with NM diameters measured to be 5–18 nm and 1600–4600 nm, the latter peak indicating some aggregation (Appendix A). The dispersions had a ZP of −17.1 ± 2.8 mV, indicating the high stability of Ag NMs in the stock dispersions (Appendix A).

The size distribution of the diluted dispersion applied for cloud exposure was confirmed to be similar to the stock dispersion (Appendix A). The hydrodynamic diameter of the Ag NMs was measured in other buffer solutions and was found to be similar in all solutions (Appendix A).

### 3.3. Deposition of Fluorescein and Ag in the Cloud System and Permeation through the Cell Barrier

Mean deposition efficiency of fluorescein reaching the surface of the cells after exposure in the VITROCELL^®^ cloud system was very consistent between the labs and calculated to 53.0% (±2.2% SD) at Lab 1 and 52.9% (±1.5% SD) at Lab 2.

The amount of Ag NMs deposited on cell culture inserts was measured by ICP-MS analysis to be 0.83 µg/cm^2^ and 6.02 µg/cm^2^, for low and high concentration, which is lower than the respective nominal concentrations 1 and 10 µg/cm^2^ (Table 1). Compared to the applied concentration, this results in a deposition efficiency of Ag of 41–56%. No Ag was detected in the blank samples (culture insert membrane without deposition).

After 24 h exposure of mono- and cocultures, a substantial amount of Ag was found on the basolateral side of the membrane, showing a permeation of Ag through the cell layers. For monocultures, the permeation was about 8% for the lower and 9% for the highest concentration of Ag NMs, corresponding to around 2 and 15 µM Ag in the basolateral media, respectively. For cocultures, the permeation was similar to monocultures for the highest concentration, but enhanced to about 14%, corresponding to 3 µM, for the lowest concentration. The maximum permeation of Ag through empty inserts without cells was estimated to be 11 µM or 7% of the high concentration, under the experimental conditions, which is at similar levels as for the mono- and coculture cell models. However, a very small amount of Ag (<0.1 µM) was in some experiments found also in the basolateral media of solvent control cells exposed to NM-300K DIS (dispersion media).

Permeation of fluorescein was in addition, measured in the coculture of A549 and EA.hy926 showing that approximately 18% of the deposited fluorescein passed through the cell barrier and was recovered on the basolateral side of the membrane separating the two cell lines in the coculture model (Figure 4). No significant differences between the treatments were observed.

### 3.4. Cell Viability

The cell viability of mono-, co-, and tricultures was measured at 20–24 h after NM-300K exposure using AB assay. The cell viability is presented relative to unexposed incubator control (NC), set to 100%. NM-300K exposure at the highest concentration significantly reduced cell viability in the monocultures compared to the PBS exposure control (Table 2 and Figure 5). A nonstatistical reduction in cell viability was seen in both the apical and basolateral cells in co- and tricultures after exposure to NM-300K at the highest concentration, and in Lab 2 also for the low concentration in tricultures, compared to the PBS exposed control. PBS exposure slightly reduced the viability of the apical cells in co- and tricultures in both labs; however, the effect was statistically significant only for tricultures in Lab 1. Cell viability after NM-300K DIS exposure was similar to in the PBS-exposed cells. No interference between NM-300K and read-out of the AB assay was found in cell-free inserts with NM-300K and AB solution (results not shown).

The effect on cell viability of different buffer solutions, PBS, PBS added CaCl_2_/MgCl_2_ (PBS+), HBSS, HBSS added CaCl_2_/MgCl_2_ (HBSS+), and DMEM cell culture medium, was investigated by exposure to cocultures of A549 and EA.hy926 at the ALI. The viability of the A549 cells was reduced compared to the incubator control, NC, for all buffer solutions tested (Appendix A). The viability of EA.hy926 cells was also reduced after exposure to the buffers, but to a lower extent than for the A549 cells. The A549 cells had the highest relative viability after exposure to PBS and HBSS+, while the EA.hy926 had the highest relative viability after exposure to HBSS and HBSS+. No statistically significant differences were found in the relative viability of cultures exposed to PBS compared to the other buffers.

### 3.5. Genotoxicity (DNA Strand Breaks and Oxidized Base Lesions)

DNA SBs and oxidized base lesions (SBs + Fpg) were measured by the enzyme-modified version of the comet assay. NM-300K induced a statistically significant increase in SBs and SBs + Fpg, measured as % DNA intensity in the tail, in the EA.hy926 cells in the cocultures at the highest concentration (with 80% DNA in tail, *p* < 0.001), compared to the two negative controls (NC, PBS). However, a slight, non-significant increase in both SBs and SBs + Fpg was seen in all models after NM-300K exposure (Figure 6). The effect of dispersant media NM-300K DIS was similar to NC (not shown). A rather high background level of damage was seen in the controls in the cocultures (NC, PBS, dispersant). The positive control for DNA SBs, H_2_O_2_, gave the expected response based on historical control (>80% SBs) (not shown). No interference between NM-300K and the performance of the comet assay was seen (results not shown).

### 3.6. Cellular Uptake of NM-300K

Uptake and intracellular localization of NM-300K was investigated by confocal microscopy and TEM after exposure to the ALI. The apical side of tricultures was investigated by confocal microscopy in unexposed cultures (Figure 7a) and after exposure to NM-300K at 10 µg/cm^2^ (Figure 7b,c). A549 cells were in higher number in the control culture (Figure 7a), and they all seemed to express pro-surfactant protein C. Rather few stained dTHP-1 cells were found. In the Ag-exposed cultures (Figure 7b), A549 cells appeared to be damaged and dTHP-1 cells highly expressed CD11b and had the appearance of activated macrophages. The phase contrast image in combination with fluorescent markers shows NMs agglomerates or aggregates in contact with A549 and dTHP-1 cells, and likely a macrophage engulfing NM-300K in the cytoplasm (Figure 7c).

Electron micrographs of the apical side of triculture are shown in Figure 8. In the cytoplasm of alveolar type II cells A549, vesicles specialized in the production of cell surfactant, called lamellar bodies (LMs), were recognized. Strong electron dense LMs and multivesicular bodies (MVB) [49] were identified in the A549 cells exposed to NM-300K, but not in unexposed tricultures (Figure 8). NM-300K appeared to be localized inside of LMs vesicles as single particles (5–20 nm) or in a small aggregate of about 100 nm (Figure 8d). No NM-300K particles were found in the endothelial cells (data not shown).

## 4. Discussion

This study aimed at testing the robustness and sensitivity, characterizing, and optimizing an advanced respiratory model built on human alveolar epithelial A549 cells and evaluating its response to aerosol exposure of silver NM-300K. In more detail, we compared the responses of A549 cells cultured at the ALI in monoculture, in coculture with EA.hy926 cells, and in triculture with EA.hy926 cells and dTHP-1, and performed an interlaboratory trial across two laboratories.

The A549 cell line is an alveolar epithelial type II cell line derived from the human lung adenocarcinoma [29]. This is currently the best characterized and used model of alveolar epithelia in in vitro studies [24,50,51]. A549 cells can partly mimic the property of the alveolar epithelium and are suitable to be used at ALI and in the VITROCELL^®^ exposure system. These cells are of alveolar origin, in contrast to alternative bronchial cell lines such as BEAS-2B, HBEC, and Calu-3.

In the alveolus, the epithelium is in close contact with the capillary endothelium. Endothelial cells are considered a secondary target for inhaled NMs, such as diesel exhaust particulate matters [52,53], and endothelial cell dysfunction is central in adverse cardiovascular disorders, including atherosclerosis, myocardial infarction, and stroke [54]. We used the lung adenocarcinoma-derived endothelial EA.hy926 cells [55,56], as they are closer to the air–blood barrier in vivo situation and thus more relevant than alternative cell lines such as HMEC-1 and hTERT-HDMEC [57,58,59].

Lung macrophages are, along with alveolar epithelial cells, the first line of defense against inhaled particles. They can express a wide range of pro- and anti-inflammatory cytokines and play a role in cellular uptake and internalization of particles [60,61]. We saw, by confocal microscopy, the presence of NM-300K inside dTHP-1 cells, which are commonly used macrophage cells [62,63] that are easy to cultivate and commonly used in ALI models [25,26,64,65,66]. In the alveolar model proposed by Klein et al. [25,52], the cell ratio between A549 and dTHP-1 does not correspond to the in situ situation where the number of macrophages is ten times less than alveolar epithelial cells [67]. We used an initial ratio of 2–4x A549/dTHP-1 cells (when at confluency), as some cells were removed during the medium removal or damaged during the detaching procedure before the seeding. Thus, the actual ratio of dTHP-1: A549 was much lower. The morphology of dTHP-1 cells was in our triculture model found to be variable and some of the cells seemed to be damaged or necrotic. The differential viability of the dTHP-1 cells in the tricultures can also be related to the culture conditions where dTHP-1 and A549 cells were sharing the same space, as well as the cell culture media composition in the tricultures, consisting of a mix of culture media optimized for each of the cell lines.

Confocal and electron microscopy confirmed that the presence and localization of all the cell types in the tricultures, with A549 and dTHP-1 cells in the apical compartment and EA.hy926 in the basolateral side of the cell culture insert. A549 and EA.hy926 cells were confluent, although some small holes or less cell dense areas could be observed, which may lead to the increased permeation of substances as shown by the permeation of fluorescein to the basolateral side of the cocultures (Figure 4). Tight junctions between the cells were seen. After 4 days in the culture, tight junction proteins were found to a large extent in the cytoplasm and to some extent in the plasma membrane of A549 cells in the triculture model. Enhanced barrier function can be obtained by extending the culturing period, as A549 cells in monocultures were expressing transcript of ZO-1 protein after two weeks at ALI [68,69,70]. Previous work by Rothen-Rutishauser et al. showed that epithelial alveolar cells formed adherent junctions and peripheral tight junctions at day 7 in culture [62,69,70,71].

When cultured in the incubator, our mono-, co-, and tricultures were moist and covered with a shiny film (results not shown). This was likely surfactant produced by the A549 cells, that were found to express surfactant proteins such as surfactant protein C [22,25,68]. Both barrier function and surfactant production may be related to cell batch, and this can influence the results, sensitivity, and reproducibility of the data obtained with the model [72,73].

Mimicking inhalation exposure in in vitro models is challenging, and the characterization of both the cell culture model and exposure system is necessary. Most of the in vitro inhalation studies have been conducted with cells in submerged conditions, which is not reflecting the air–blood barrier in the lungs. Further, the cell medium can influence the properties of the NMs and thereby give a non-realistic judgment regarding the hazardous potential of NMs and other compounds. Enhanced predictiveness for human effects after inhalation exposure is likely to be obtained with cells cultivated at the ALI, and multicellular models are better mirroring organ-like structures. Thus, to compare sensitivity to NM exposure when adding complexity to the ALI model, mono-, co-, and tricultures were exposed in the VITROCELL^®^ system, where the NM suspension or control solution is nebulized to generate an aerosol which over a few minutes deposits onto the cells. The deposited concentrations of Ag in the VITROCELL^®^ system were found to be 0.8 µg/cm^2^ and 6.0 µg/cm^2^, which was lower than expected. We measured by ICP-MS an Ag content of the stock concentration of NM-300K at 7.2 mg/mL, which was lower than the nominal concentration at 10 mg/mL The presence of 3.6% dissolved species in the stock dispersion was as expected [36,38,74] and the very low concentration of Ag (1.8 ng/mL) found in the dispersion media was likely caused by contamination during preparation/sonication. The deposition of Ag was measured only at Lab 1; however, as the deposition efficiency of Ag was like that of fluorescein, and the same protocols were used, it is likely that the results would be similar in both laboratories. Quartz crystal microbalance (QCM) has been reported as a precise and sensitive device for quasi-real-time NM dosimetry for the VITROCELL^®^ system [39]. We experienced high inter-experimental variations with the QCM (results not shown) and thus used other methods for the determination of NM deposition. The uptake of NM-300K after aerosol exposure was shown in both A549 and dTHP-1 cells on the apical side. It has been demonstrated that alveolar type II cells can internalize NMs [4]. Ag NMs also appeared inside of dTHP-1 cells, recognizable by the absence of lamellar bodies (LMs). The internalization of silver nanoparticles in A549 and THP1 cells has previously been shown in several investigations [75,76]. LMs are typical of the epithelial alveolar type II cells, and the electron micrographs showed the presence of LMs in A549 cells. Several studies reported an altered quantity of LMs in A549 cells after NMs exposure [60,77,78]. Ag-NMs were internalized in dTHP-1 cells as single particles, and as agglomerates up to 100 nm size (Figure 8d). By confocal microscopy analysis, dTHP-1 cells appeared to have a phagosome-like morphology typical of activated macrophages [79]. Moreover, we observed that the dTHP-1 cells exposed to the highest NM concentration phagocytosed particles which were visible inside of their cytoplasm.

NM-300K was chosen as the test substance for characterization of the different models due to known cytotoxic properties [33,35], and the NM-300K stock dispersion was found to be endotoxin-free, measured by both LAL and HEK293 assays. The large surface area and other surface properties of NMs make them susceptible to endotoxin contamination during the synthesis process. Endotoxin contamination may bias the results of toxicity testing by false positive or negative results if not controlled for [80].

The cellular viability of the lung cells was affected by NM-300K exposure, but also by buffer exposure. Viability was reduced after NM-300K exposure at the highest concentration in monoculture (*p* < 0.01), while in cocultures and tricultures no significant decrease in cell viability was observed. The effect on viability caused by NM-300K was strongest at Lab 2, which may be related to the measured smaller diameter of the NMs used for exposure of the cells. The NM sizes were relatively constant for all dispersions within each laboratory, both measured in stock dispersion and in stock dispersion diluted in physiological buffers. All dispersions showed, by DLS measurements, multiple peaks for the size distribution of the NMs in the dispersion, but the dispersed NMs prepared at Lab 1 were larger than those prepared at Lab 2. To test if the difference in size was related to the two different DLS instruments, three dispersions prepared at Lab 1 were tested in both laboratories, but no significant differences were found (results not shown). This points to the importance of measuring the size, size distribution, and stability of all dispersions used for exposure of cells and toxicity measurements, as the toxic effect is strongly dependent upon physicochemical properties, such as size, of the NMs. In the AB results from both laboratories, we saw higher inter-experimental variation compared to studies with A549 cells in submerged conditions [35,73], which may be related to the complexity of the ALI cultures and the experimental conditions. However, the effect seen on the viability of A549 monocultures after NM-300K exposure at the highest concentration was similar to submerged experiments with A549 [35].

The monocultures were more sensitive to the induction of cytotoxicity by NM-300K compared with the more complex models. However, the control exposure with PBS and HBSS with and without CaCl_2_ and MgCl_2_ reduced the viability of the apical cells in both co- and tricultures, but not in monocultures. This effect was strongest in Lab 1. The high sensitivity of the co- and tricultures to aerosol exposure in general needs to be investigated further. The effects of PBS exposure on the cell viability of the A549 cells could be influenced by less contact with the basolateral media due to the introduction of EA.hy926 cells. High confluency of the EA.hy926 cells would limit more strongly the access of the A549 cells to the basolateral media and constitute an important factor to be aware of for the preparation of the most optimal ALI model. The use of PBS diluted in water (1:10) can be an option to improve the viability of the cells in the negative control (Appendix A), as this will reduce the salts deposited on top of the cells when the water evaporates, while still making a dense aerosol [26]. The choice of solvent will also depend on the solubility of the test particles. Exposure to PBS did not induce an increase in DNA damage; however, the highest concentration of NM-300K induced DNA SBs in EA.hy926 cells in cocultures. In tricultures, no effect of NM-300K was seen on DNA damage in EA.hy926 cells. This might be linked to the potential uptake of NMs by the dTHP-1 cells as previously shown [75,81] making the material less available for cell exposure. One can even speculate if the exposed THP1 cells trigger an inflammatory reaction that can promote DNA repair [82]. The cocultures also had a slightly higher background level of DNA damage compared with mono- and tricultures.

The effect seen on the EA.hy926 cells in cocultures can be caused by signals from the apical side or from the permeation of Ag ions or Ag particles through the cell culture barrier into the basolateral side. In the coculture model, around 3 µM and 15 µM Ag for the lower and higher concentration, respectively, was measured in the basolateral medium after exposure, and it is unknown if this was in form of Ag NMs or dissolved species. Reduced viability and increased DNA damage have previously been reported after exposure to Ag NMs in standard submerged cell cultures [83,84,85,86] at concentrations in the same range as the dissolved Ag we detected. The permeation of Ag was lower than that of fluorescein in the cocultures of A549 and EA.hy926 cells, which was expected as fluorescein is a water-soluble molecule. Permeation of Ag NMs or dissolved species into the basolateral medium is more likely to be limited by cell adhesion/uptake or protein binding on cells on either side of the membrane, as Ag has low solubility in physiological medium and high affinity to thiol groups [87,88,89]. The maximum permeation of Ag through empty inserts without cells was estimated to be around 20%, showing that a substantial amount of the Ag (NMs or dissolved species) was remaining in the apical side or in the insert pores.

Interference between the tested NM and the assay is commonly seen with metallic NMs. No interference was found between NM-300K and the AB and comet assays (results not shown). In contrast, interference of NM-300K with the lactate dehydrogenase (LDH) assay was found after analysis of the basolateral medium of the exposed ALI cultures (results not shown), in line with previously reported data [26,90].

## 5. Conclusions

The advanced 3D lung model with aerosol exposure at the ALI in the VITROCELL^®^ Cloud chamber was shown to be a promising in vitro model for hazard identification and characterization of NMs in relation to inhalation exposure. Thus, this work is supporting the ongoing effort to implement NAMs and advanced in vitro methods for regulatory purposes and to replace animal studies, in compliance with the 3Rs. This respiratory model was shown to be compatible with different endpoints—cytotoxicity by the AlamarBlue assay and genotoxicity by the enzyme-modified version of the comet assay. NM-300K at the highest concentration tested was shown to be cytotoxic in the monocultures, and induced DNA strand breaks and oxidized base lesions in the EA.hy926 cells in cocultures. No interference of NM-300K with the AlamarBlue and comet assay was detected; however, interference was detected for the LDH assay. Ag NMs were found to be taken up by the A549 and THP-1 cells, and the deposition efficiency of Ag was measured by ICP-MS to be 41–56%, which is comparable to the deposition efficiency of fluorescein (53%). A549 cells were found to express surfactant protein, as well as tight junctions. However, the barrier was not complete, as Ag was found in the basolateral medium, at a comparable level for both mono- and cocultures, and fluorescein in a substantial amount in the cocultures. Adding complexity to the model, going from monocultures of lung epithelial cells to cocultures with endothelial cells or tricultures by further adding immune cells, changed the sensitivity for exposure to both NM-300K and PBS. This points to the importance of the development and characterization of advanced multicellular in vitro models when moving from monolayer cultures of human cells into 3D and from one type of cell into advanced culture containing several cell types relevant to the organ of interest, as sensitivity and effects can change. There will always be a question about which is the best model to predict human effects. The application of human cells is considered to be advantageous, and advanced 3D models, which are closer to tissue and organ structure, should be more reliable for toxicity testing in vitro for human hazard assessment. More complex models are more challenging to work with and will introduce more variation in the data, as expected and shown by our inter-laboratory comparison study. Thus, detailed and optimized protocols are important to increase reproducibility and robustness. For NGRA, models that are well tested, robust, characterized, and standardized for validation are needed. We, therefore, performed an interlaboratory trial across two labs to characterize the advanced models, test their robustness, and optimize the protocol, and showed that adding realistic complexity by including several cell types changed the outcome of the toxicity testing.

## Figures and Tables

**Figure 1 nanomaterials-12-02609-f001:**
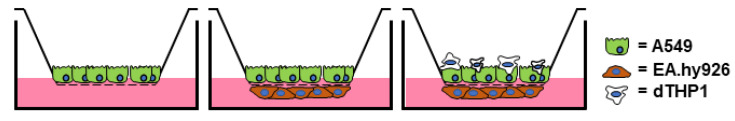
Scheme of mono-, co-, and triculture lung models of A549, EA.hy926, and differentiated THP-1 (dTHP-1) cells cultivated at the air–liquid interface (ALI) on membrane inserts.

**Figure 2 nanomaterials-12-02609-f002:**
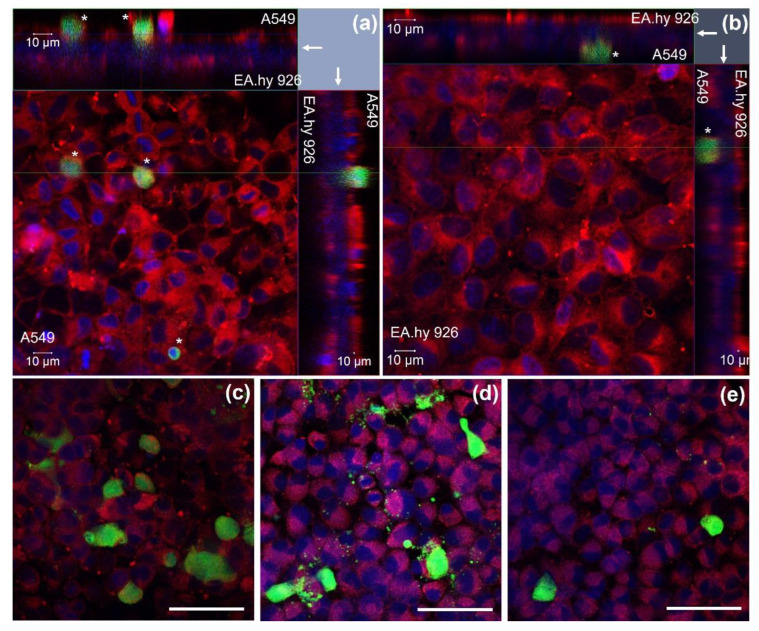
Triculture model investigated by confocal microscopy. (**a**,**b**) Z-stack image series (2D x-y view and respective side views) showing the distribution of A549, EA.hy926, and dTHP-1 (*, green) cells on the opposite sides of a transwell insert (arrow). (**a**) The 2D x-y view from the A549 and dTHP-1 side (z-stack thickness 44.5 µm). (**b**) The 2D x-y view from the EA.hy926 side (z-stack thickness 27.5 µm). (**c**–**e**) dTHP-1 cells in different morphologies on top of the A549 cells. Red: cellular membranes stained with Cell Mask red dye, blue: nuclei counterstained with DAPI, green: dTHP-1 cells stained with Cell tracker green dye. Magnification: 40×. Scale bar 10 µm (**a**,**b**) and 50 µm (**c**–**e**). dTHP-1: differentiated THP-1.

**Figure 3 nanomaterials-12-02609-f003:**
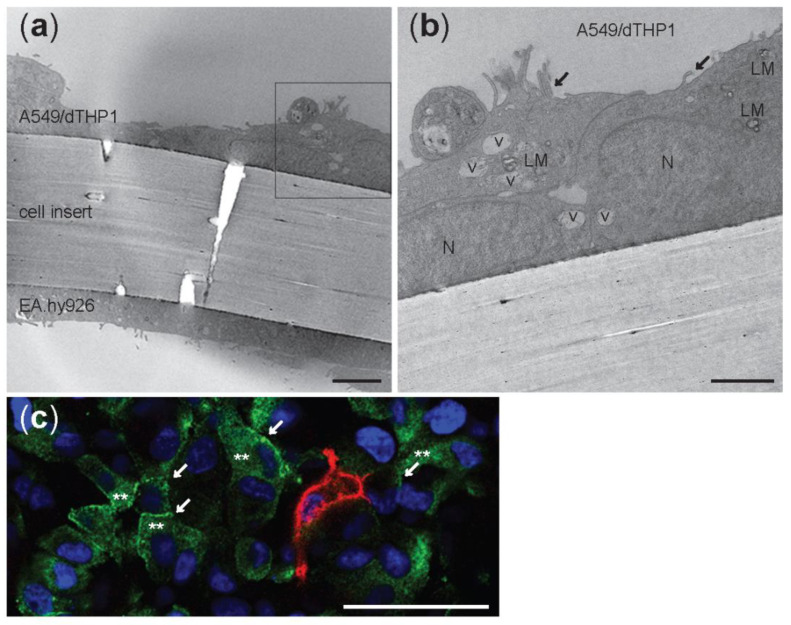
Transmission electron microscopy micrographs (**a**,**b**) and confocal picture © of unexposed triculture with A549 and dTHP-1 cells at the apical side and EA.hy926 cells at the basolateral side of a cell insert with 1 µm pore size membrane. LM: lamellar bodies; v: vesicles; N: nuclei; black arrows: microvilli. (**a**) Scale bar: 5 µm. (**b**) Scale bar: 1 µm. (**c**) Confocal image of the apical side. A549 were stained with ZO-1 antibody (green) and differentiated THP-1 with CD11b (red). White arrows: tight junctions. **: cytoplasmic ZO-1. Nuclei are stained in blue (DAPI). Scale bar: 50 µm; Magnification: 40×.

**Figure 4 nanomaterials-12-02609-f004:**
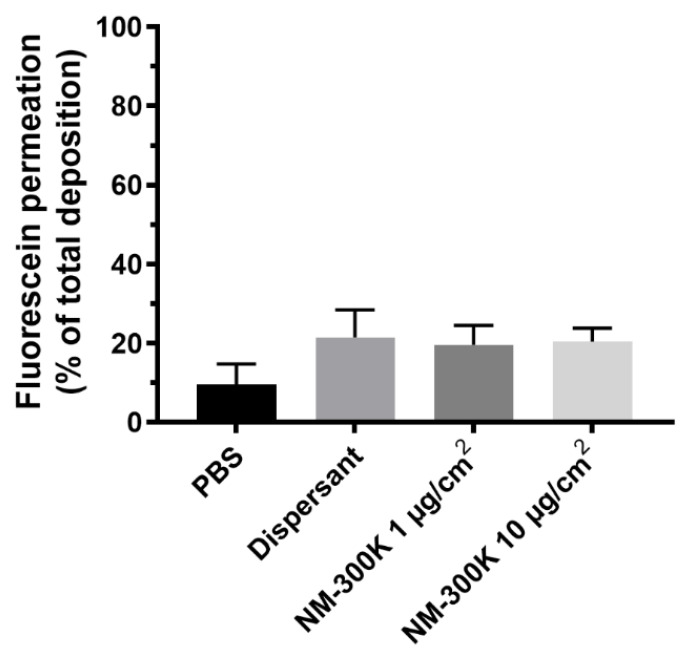
Permeation of fluorescein through the cellular layers. Shown is mean recovery of fluorescein ± SD in the basolateral compartment relative to the total deposition of fluorescein at the apical side of the membrane after exposure of cocultures in the cloud system to phosphate buffered saline (PBS), dispersant NM-300K DIS, or NM-300K at 1 or 10 µg/cm^2^. The results are based on 4 technical replicates in *n* = 3 independent experiments. No statistically significant difference was seen between PBS treatment and the other samples, analyzed by ordinary one-way ANOVA with multiple comparisons, post-test Dunnett’s, *p* > 0.05. SD: standard deviation.

**Figure 5 nanomaterials-12-02609-f005:**
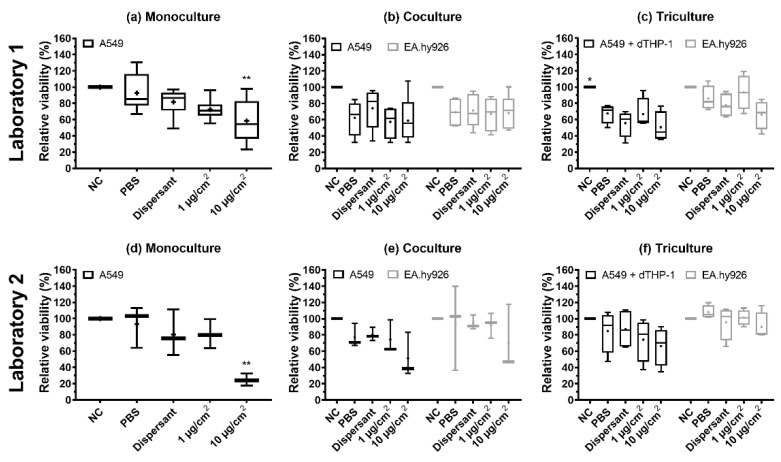
Relative viability measured by the AlamarBlue assay after exposure of monocultures (**a**,**d**), cocultures (**b**,**e**), and tricultures (**c**,**f**) of A549, EA.hy926, and dTHP-1 cells to NM-300K and control solutions at the air–liquid interface (ALI). Experiments were performed at Lab 1 (**a**–**c**) and Lab 2 (**d**–**f**) for comparison. A reduction in viability was seen after exposure to control solutions and NM-300K at ALI. Results are normalized against negative incubator control (NC, set to 100%) and presented as boxplots with mean (+), median/50th percentile (line), 25th and 75th percentiles (box), and minimum and maximum values (whiskers). A total of (**a**) *n* = 6–9, (**b**) *n* = 4–5, (**c**) *n* = 4, (**d**,**e**) *n* = 3, and (**f**) *n* = 4 independent experiments with each 1–2 replica cell culture inserts were performed. Statistically significant differences compared to PBS control were analyzed by ordinary one-way ANOVA with multiple comparisons post-test Dunnett´s, and are indicated by * *p* < 0.5, ** *p* < 0.1. NC: negative control (incubator control), PBS: phosphate buffered saline, dispersant: NM-300K dispersion medium (NM-300K DIS); dTHP-1, differentiated THP-1 cells.

**Figure 6 nanomaterials-12-02609-f006:**
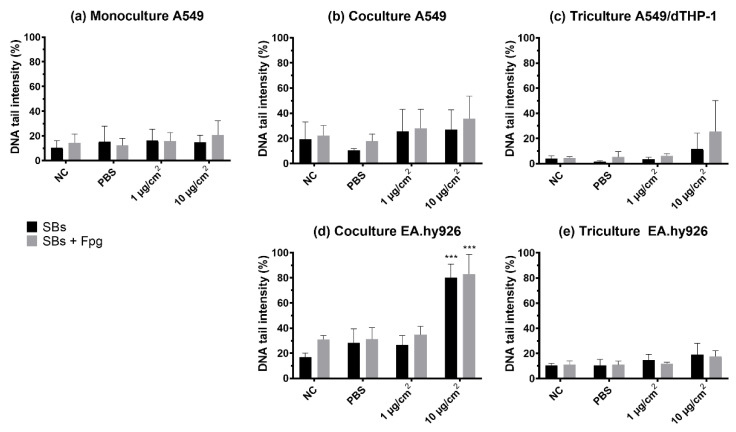
DNA strand breaks and oxidized DNA lesions, measured as DNA tail intensity, by the comet assay with Fpg in cells exposed at the air–liquid interface (ALI) to NM-300K. Cells were cultured as (**a**) monocultures, (**b**,**d**) cocultures, and (**c**,**e**) tricultures. Results are presented as mean of median ± SD of (**a**) *n* = 3–5, (**b**–**e**) *n* = 3 independent experiments. From each experiment, the median DNA tail intensity (%) was calculated from 50 cells per 2–6 gels from 1–2 cell culture inserts. Significantly different effects on DNA damage compared to PBS control were analyzed by ordinary one-way ANOVA followed by Dunnett’s post-hoc test (*** *p* < 0.001). SBs: strand breaks, NC: negative control, PBS: phosphate buffered saline, SD: standard deviation.

**Figure 7 nanomaterials-12-02609-f007:**
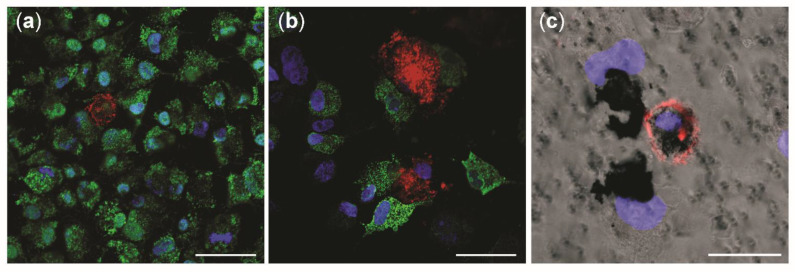
Confocal microscopy investigation of tricultures with and without NM-300K exposure. Confocal pictures show the apical side of triculture model in (**a**) negative incubator control, and (**b**,**c**) after exposure to NM-300K at 10 µg/cm^2^. A549 cells were stained with pro-surfactant protein C (green) and dTHP-1 are marked with CD11b (red). DNA is stained with DAPI (blue). (**c**) Phase contrast picture combined with immunofluorescence staining. (**a**–**c**) Magnification: 40×. Scale bar: 50 µm (**a**,**b**); 20 µm (**c**).

**Figure 8 nanomaterials-12-02609-f008:**
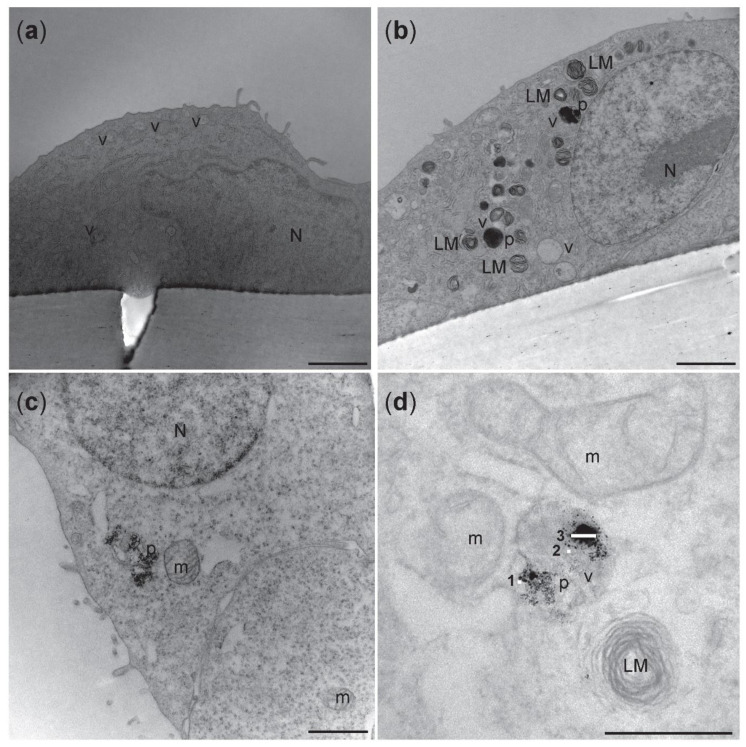
Representative transmission electron micrographs of A549 cells on the apical side of the triculture model. (**a**) Unexposed cells from incubator control (NC) showed no electron dense vacuoles. (**b**) Electron dense vacuoles-like structure and lamellar bodies were seen in cells exposed to NM-300K 10 µg/cm^2^. (**c**,**d**) Higher magnification of the cells exposed to NM-300K 10 µg/cm^2^ showed that NM-300K were found in the cell cytoplasm (**c**) or inside of a vacuole (**d**). (**d**) NM-300K were found as single particles (5–20 nm) (1,2) or in small aggregate of 100 nm (3). Scalebar: 2 µm (**a**,**b**); 1 µm (**c**); 500 nm (**d**). N: nucleus, LM: lamellar bodies, v: vacuoles-like structure, p: NM-300K nanoparticles, m: mitochondria.

**Table 1 nanomaterials-12-02609-t001:** Deposition and permeation of Ag in mono- and cocultures exposed to an aerosol of NM-300K. Control was exposed to NM-300K DIS. Results are presented as mean with SD. Deposition was measured in 2 or 4 independent experiments with 2–3 replica inserts in each experiment. Permeation was measured in 2–3 independent experiments, each with 1–2 replica inserts. Maximum permeation of NM-300K at high concentration through empty inserts was 7% (11 µM, *n* = 2). * Control was insert membranes without Ag deposition. LOD, limit of detection; SD, standard deviation.

	Unit	Solvent Control	Low Concentration	High Concentration
Nominal deposited concentration	µg/cm^2^	0	1	10
Measured deposited concentration	µg/cm^2^	<LOD *	0.83 ± 0.05 (*n* = 2)	6.02 ± 0.84 (*n* = 4)
Deposition efficiency	% of nebulized	-	56 (*n* = 2)	41 (*n* = 4)
Permeation of Ag through monoculture cell model	µM (% of deposited)	0.069 ± 0.025 (*n* = 2)	1.9 ± 0.56 (*n* = 2) (8%)	14.5 ± 1.4 (*n* = 3) (9%)
Permeation of Ag through coculture cell model	µM (% of deposited)	0.030 ± 0.017 (*n* = 2)	3.3 ± 0.5 (*n* = 2) (14%)	15.4 ± 1.2 (*n* = 2) (9%)

**Table 2 nanomaterials-12-02609-t002:** Cell viability (%) relative to incubator control (set to 100%) measured by the AlamarBlue assay after exposure of monocultures, cocultures, and tricultures of A549, EA.hy926, and dTHP-1 cells to NM-300K (1 and 10 µg/cm^2^) or control solutions (PBS, dispersant (NM-300K DIS)) at the air–liquid interface (ALI). Results are presented as mean with standard deviation from a total of *n* = 6–9 (Lab 1), *n* = 3 (Lab 2) for monocultures, for cocultures *n* = 4–5 (Lab 1), *n* = 3 (Lab 2), and for tricultures *n* = 4 independent experiments with each 1–2 replica cell culture inserts, corresponding to results in Figure 5. Statistically significant differences compared to PBS or NC incubator controls were analyzed by one-way ANOVA with multiple comparisons post-test Dunnett’s and are indicated by ^a^ *p* < 0.5 compared to PBS, and ^b^ *p* < 0.05 compared to NC. NC: negative control, PBS: phosphate buffered saline, dTHP-1: differentiated THP-1 cells.

		Relative Cell Viability (%)					
		Monoculture		Coculture		Triculture	
Cells	Treatment	Lab 1	Lab 2	Lab 1	Lab 2	Lab 1	Lab 2
A549/ A549- dTHP-1	NC	100 ± 0	100 ± 0	100 ± 0	100 ± 0	100 ± 0 ^a^	100 ± 0
	PBS	93 ± 23	93 ± 26	62 ± 22	77 ± 15	68 ± 12 ^b^	85 ± 26
	Dispersant	82 ± 17	81 ± 28	74 ± 25	80 ± 8	56 ± 17 ^b^	87 ± 24
	Ag 1 µg/cm^2^	73 ± 13 ^b^	81 ± 18	58 ± 20 ^b^	75 ± 21	67 ± 19 ^b^	74 ± 26
	Ag 10 µg/cm^2^	59 ± 26 ^ab^	25 ± 8 ^ab^	59 ± 29 ^b^	51 ± 28 ^b^	51 ± 19 ^b^	66 ± 23
EA.hy926	NC			100 ± 0	100 ± 0	100 ± 0	100 ± 0
	PBS			69 ± 19	93 ± 52	86 ± 15	108 ± 8
	Dispersant			71 ± 21	94 ± 9	78 ± 15	96 ± 21
	Ag 1 µg/cm^2^			67 ± 21 ^b^	92 ± 16	94 ± 21	101 ± 10
	Ag 10 µg/cm^2^			68 ± 21 ^b^	70 ± 41	66 ± 18	90 ± 18

## Data Availability

Data are available from the researchers on request.

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
