# Peer review of "Advanced Respiratory Models for Hazard Assessment of Nanomaterials—Performance of Mono-, Co- and Tricultures"

_nanomaterials, 2022, doi:10.3390/nano12152609_

Round 1

Reviewer 1 Report

General comment

In this study authors performed an interlaboratory trial across two laboratories in order to evaluate the performances of mono-, co-, and triple cultures of A549 epithelial cells, A549 co-cultured with EA.hy926 endothelial cells, and triple culture including macrophages differentiate THP-1 cells following exposures to silver nanoparticles NM-300K at the Air-Liquid Interface.

Overall the work addresses important topics that are highly relevant to improve available test systems to assess particle toxicity. The rationale for the work is well laid out, the hypothesis is well described and the choice of endpoints to study is well supported by the literature. The paper is well written and easy to read, however there are major flaws which should be taken into consideration before considering the paper for publication.

Major comments:

1)      Table 2 and Figure 5: Please associate an error to the NC values since 1-2 replica per independent experiments were made. This will impact statistical analysis.

2)      Table 2 and Figure 5: Is the reduction in viability observed following control solutions in both coculture and triculture impacting the sensitivity of the model systems in detecting NM-300K induced effects? Please discuss it also considering the confluency of EA.hy926 cells (Figure 2 and 3) which could impact the A549 access to the basolateral cell culture medium.

3)      Figure 6: is the relative viability of PBS control around 50% interfering with Comet data? Since the Comet was performed after the Alamar blue assay and cells were washed twice, authors should consider that DNA damage was evaluated only in remaining live cells. This is the case also for Ag treated cells. Please, provide at least n=3 for Fpg experiments on Ag 10 µg/cm2 exposures.

4)      Discussion, lines 776-778: Authors state that no complete barrier function was found in mono- and co-cultures due to the penetration of fluorescein in the basolateral side. Are there differences between mono- and co-cultures? (No data provided for the monocultures)

5)      Discussion, lines 778-782: authors state that tight junctions were observed in triple cultures since ZO-1 protein was found in the plasma membrane (only in few cells) and in the cytoplasm (not in all cells). However, no continuous circumferential localisation of ZO-1 can be observed which is a characteristic of cells forming tight junctions (DOI: 10.1183/09031936.00065809). Furthermore, authors state that their observation is in accordance with Rothen-Rutishauer and Elbert papers where authors could not find any ZO-1 expression in their experiments. Please consider to revise these statements according to the obtained results and cited literature.

6)      Discussion, lines 860-865. Authors observed DNA damage on EA.hy926 cells in co-cultures but not in triple cultures linking to the potential dTHP-1 NM-300K uptake such difference. What about Ag permeation in the triple cultures? No significant differences were noticed between mono- and co-cultures (Table 1). What about Ag permeation in triple culture?

Minor comments

Line 722: please use LMs instead of LBs for lamellar bodies or define as LBs the lamellar bodies.

Author Response

Dear reviewer, thanks for your valuable comments on our paper. Please see the attachment for our replies. 

Reviewer 2 Report

The paper of Camassa and Elje describes a very well designed experiment aiming to test the robustness of an in vitro 3D lung model in testing  the nanomaterials hazard. The tests have been done in two different laboratories and the results have been very similar. The results have demonstrated the utility of this model as an alternative to in vivo studies. The paper is very well written and can be published in Nanomaterials journal.

Author Response

Dear reviewer, thanks for your time to review our paper.

Round 2

Reviewer 1 Report

Thank you for revising the manuscript according to the reviewer comments.